# Immunogenicity of the Conjugate Meningococcal ACWY-TT Vaccine in Children and Adolescents Living with HIV

**DOI:** 10.3390/microorganisms12010030

**Published:** 2023-12-23

**Authors:** Arantxa Berzosa, Sara Guillen, Cristina Epalza, Luis Escosa, Maria Luisa Navarro, Luis M. Prieto, Talía Sainz, Santiago Jimenez de Ory, Marina Montes, Raquel Abad, Julio A. Vázquez, Irene Serrano García, José Tomás Ramos-Amador

**Affiliations:** 1Pediatric Infectious Diseases Unit, Department of Paediatrics, Clínico San Carlos Hospital, 28040 Madrid, Spain; 2Health Research Institute of the Clínico San Carlos Hospital (IdISSC), 28040 Madrid, Spain; iserrag01@gmail.com; 3Centro de Investigación Biomédica en Red de Enfermedades Infecciosas (CIBERINFEC), 28029 Madrid, Spain; sguillenmartin@hotmail.com (S.G.); luisescosa1983@gmail.com (L.E.); marisa.navarro.gomez@gmail.com (M.L.N.); josetora@ucm.es (J.T.R.-A.); 4Pediatric Infectious Diseases Unit, Department of Pediatrics, Hospital de Getafe, 28905 Madrid, Spain; 5Pediatric Infectious Diseases Unit, Department of Pediatrics, Hospital Universitario 12 de Octubre, 28041 Madrid, Spain; crepalza@hotmail.com (C.E.); prieto_tatoluis_manuel@hotmail.com (L.M.P.); 6Department of Pediatrics, Infectious and Tropical Diseases, Pediatrics, La Paz University Hospital, 28046 Madrid, Spain; 7Hospital La Paz Institute for Health Research (Idipaz), 28029 Madrid, Spain; 8Department of Pediatrics and IISGM, Gregorio Marañón Hospital, 28007 Madrid, Spain; 9Department of Pediatrics, Universidad Complutense de Madrid (UCM), 28040 Madrid, Spain; 10Department of Pediatrics, Universidad Autonoma de Madrid (UAM), 28029 Madrid, Spain; 11Health Research Institute of the Gregorio Marañón Hospital (IisGM), 28009 Madrid, Spain; 12Neisseria, Listeria and Bordetella Unit, Reference and Research Laboratory for Vaccine Preventable Bacterial Diseases, National Centre for Microbiology, Instituto de Salud Carlos III, 28029 Madrid, Spain; marina.montesm@isciii.es (M.M.); rabad@isciii.es (R.A.); jvazquez@isciii.es (J.A.V.); 13Pediatric Infectious Diseases Unit, Head of Department of Pediatrics, Clínico San Carlos Hospital, 28040 Madrid, Spain

**Keywords:** HIV, meningococcal infection, Men ACWY-TT vaccines, vaccine response

## Abstract

Background: Children and adolescents living with HIV (CALHIV) are at high risk of meningococcal infections and may present lower immune responses to vaccines. The objectives of this study were to assess the immunogenicity of the quadrivalent Men ACWY-TT vaccine (Nimenrix^®^) in CALHIV after a two-dose schedule and to describe possible HIV-related factors that may affect the immunogenic response. Methods: A multicenter prospective study was designed, including CALHIV followed in five hospitals in Madrid, between 2019 and 2021. Two doses of the Men ACWY-TT vaccine were administered. Serum bactericidal antibody (SBA) assays using rabbit complement (rSBA) against serogroups C, W, and Y were used to determine seroprotection and vaccine response (the proportion achieving a putative protective titer of ≥eight or a ≥four-fold rise in titer from baseline). Serum was collected at baseline, and at 3 and 12 months after vaccination. Results: There were 29 CALHIV included, 76% of whom were perinatally infected. All were receiving TAR and presented a good immunovirological and clinical status overall. At baseline, 45% of CALHIV had seroprotective titers to at least one serogroup, with individual seroprotection rates of 24%, 28%, and 32% against C, W, and Y, respectively. After a two-dose schedule, vaccine response was 83% for each serogroup, eliciting a vaccine response to all serogroups in 69% of them. One year after vaccination, 75% of CALHIV maintained seroprotective titers against the C serogroup, and 96% against W and Y. None of the HIV-related characteristics analyzed could predict vaccine response or antibody duration. Conclusions: CALHIV who received effective TAR and presented a good immuno-virological situation achieved an appropriate vaccine response after two doses of the Men ACWY-TT vaccine, and antibody-mediated protection against serogroups C, W, and Y was maintained in more than 70% of the patients one year after vaccination.

## 1. Introduction

Invasive meningococcal disease (IMD) is caused by *Neisseria meningitidis,* a Gram-negative diplococcus. Of the capsular type, there are 12 serogroups described [1], with the most frequently isolated in Europe being B, W, C, and Y. They are responsible for most cases of invasive disease. Although it is not a common disease, it is associated with high morbidity and mortality. In our setting, it presents an average mortality rate of around 10%. Severe sequelae are described in up to 30% of cases [2]. The incidence of IMD appears to be higher in children under 5 years of age and in the elderly, but adolescence poses an additional risk of nasopharyngeal carriage and transmission [1].

People living with HIV (PLHIV) are at increased risk of meningococcal infections. Surveillance studies in Atlanta, prior to the implementation of conjugate immunization programs, found an incidence of meningococcal disease in PLHIV 24 times higher than in non-HIV-infected people [3]. A more recent study, performed in the era of highly active antiretroviral therapy also in the United States, reported a 10-fold increased risk of IMD in PLHIV [4]. Other series from South Africa have also reported not only an 11-fold increased risk in this population but also a higher severity, with mortality around 20% [5]. In England, another series described a five-fold higher incidence of IMD in PLHIV [6]. In this series, Simmons et al. reported that the risk of IMD caused by groups C, W, or Y in PLHIV aged 16–64 years was increased 23-fold compared with that in adults uninfected with HIV. This risk has been described as inversely proportional to immunosuppression status, with higher incidence in PLHIV with less than 200 CD4+ cell/mm^3^ [7].

The most effective preventive measure for IMD is meningococcal conjugate vaccination. There are four meningococcal-conjugate quadrivalent vaccines against serogroups A, C, W, and Y, three of which are approved in Europe. Among those approved in Europe in recent years, one is conjugated to the diphtheria CRM197 protein carrier (Menveo^®^) and two to the tetanus toxoid protein carrier (Nimenrix^®^ and, more recently, MenQuadfi^®^). In the United States, the Men ACWY quadrivalent vaccine, conjugated to the diphtheria toxoid protein carrier (Menactra^®^), is also approved. The introduction of these vaccines has led to significant reductions in the number of meningococcal invasive disease cases due to both the direct protection of those patients vaccinated and to a herd immunity effect through the impact on nasopharyngeal carriage and secondary transmission.

These meningococcal-conjugate quadrivalent vaccines are safe and immunogenic in children and adolescents. However, previous studies have demonstrated that children and adolescents living with HIV (CALHIV) have a lower immune response to common vaccines when compared with healthy subjects [8]. Data regarding meningococcal immunogenicity were published by Frota AC et al., who suggested that a single dose of the Men C conjugate vaccine was poorly immunogenic in CALHIV aged 2 to 18 years. They found that seroconversion in CALHIV was significantly lower (30%) than in non-HIV children and adolescents (76%) after one dose of the vaccine [9]. Since this population may have impaired vaccine response, a two-dose booster schedule of the Men ACWY vaccine, conjugated to the tetanus toxoid protein carrier (Men ACWY-TT), might be needed for PLHIV, as recommended by the Advisory Committee on Immunization Practices of the United States [10]. A recent open-label clinical trial in the UK, on young adults living with HIV, showed that more than 90% of individuals vaccinated simultaneously with two doses of both a quadrivalent conjugate polysaccharide Men ACWY vaccine (MenACWY-CRM197) and with a four-component recombinant protein-based Men B vaccine (4CMenB) achieved a short-term protective titer for the four serogroups included in the Men ACWY vaccine [11].

The main aim of our study was to assess the immunogenicity of the quadrivalent Men ACWY-TT vaccine first licensed in Europe (Nimenrix^®^) in CALHIV after a two-dose schedule. Secondly, we aimed to describe possible HIV-related factors that may affect the immunogenic response.

## 2. Methods

### 2.1. Study Design and Patients

A multicenter prospective study was conducted in Madrid, including children and adolescents living with HIV (CALHIV) and followed in five pediatric HIV clinics belonging to the Spanish Paediatric HIV Cohort (CoRISpe). CoRISpe was set up to collect data from HIV-infected children throughout Spain retrospectively since 1995 and prospectively from 2008 [12]. They were included between January 2019 and December 2021.

The quadrivalent Men ACWY-TT vaccine (Nimenrix^®^) was approved for use and financed in the Madrid region in Spain in 2019 for adolescents and people with underlying conditions, including HIV infection. It was at that moment that CALHIV were invited to participate in this study.

All patients had to meet the inclusion criteria: a confirmed diagnosis of HIV infection, being older than 2 years at the time of the first dose of the Men ACWY-TT vaccine, absence of concomitant vaccination or having received any live-attenuated virus vaccine in the previous month, and absence of previous meningococcal infection or previous vaccination with a Men ACWY vaccine. Those who were younger than two years, had previous vaccination against *Neisseria meningitidis* ACWY, history of infection or receipt of any vaccination in the previous month were excluded. A history of vaccination with any Men B protein-based vaccine or Men C conjugate vaccine in the past was not an exclusion criterion.

Vaccination was administered according to the data sheet: two doses of Men ACWY-TT, deep intramuscular injection, 2 months apart. Vaccination was performed by qualified health personnel (either a physician or nursing staff), and patients were required to remain for half an hour under supervision to closely monitor any potential adverse event. Additionally, they were required to promptly report any reactions to or side effects of the vaccine within the first 48 h.

The blood samples were scheduled to be collected at baseline before the immunization process, just before administering the first Men ACWY-TT dose (T0); approximately one month after the second dose (T1); and, finally, twelve months after the first dose (T2). Sera were meticulously frozen immediately to ensure their preservation for further analysis.

### 2.2. Data and Laboratory Testing

Epidemiological data (including age, sex, and country of birth) and HIV-related characteristics (antiretroviral treatment (ART) received, CD4+ cell count and CD4+ nadir and viral load) were obtained from the CoRISpe database at the moment of the inclusion in the study. Vaccination history against Men B or Men C (conjugate vaccine) was registered from the centralized electronic medical records of vaccination in the Madrid region.

Vaccine immunogenicity was assessed through serum bactericidal antibody (SBA) assays using rabbit complement (rSBA) against a strain from each of serogroups C, W, and Y at the National Meningococcal Reference Laboratory (National Centre for Microbiology, Instituto de Salud Carlos III, Majadahonda, Madrid). Serially diluted serum samples were incubated with the serogroups’ specific meningococcal strains and baby rabbit complement, and rSBA titers were defined as the highest dilution at which ≥50% killing occurred [13].

An rSBA titer of ≥1:8 was used as a serological correlate of protection. Vaccine response was defined as a postvaccination rSBA titer of ≥1:32 in initially seronegative subjects. For subjects with prevaccination rSBA titers ≥1:8, vaccine seroresponse was defined as a ≥4-fold increase in titers from baseline [14]. A prevaccination rSBA titer of ≥1:8 was considered seroprotective.

### 2.3. Ethics

This study was officially approved by the Ethics Committee of the Biomedical Research Institute of the coordinating center, Clínico San Carlos Hospital, under the internal code: 20/136-E. Furthermore, it was approved by all the Institutional Review Boards of the participating centers (La Paz University Hospital, Doce de Octubre University Hospital, Gregorio Marañón University Hospital and Getafe University Hospital).

The investigators followed the applicable ethical and legal standards (the Declaration of Helsinki) and the standards of good practice in human research. Participants or their parents or legal representatives signed an informed consent before inclusion in the study. Also, all patients ≥ 12 years of age at the moment of inclusion signed an assent form. Data were anonymized to ensure confidentiality.

### 2.4. Statistical Analysis

Demographic data and baseline characteristics were analyzed. Qualitative variables were summarized using their frequency distribution. The continuous non-normally distributed variables were summarized using the median and interquartile range (IQR: P 25–P 75). Pearson’s correlation coefficient was used to compare quantitative variables, and qualitative variables were compared using the chi-square test or Fisher’s exact test when appropriate. If the qualitative and quantitative variables under study were independent, the nonparametric test, the Mann–Whitney *t*-test, or the Kruskal–Wallis test (if there were more than two categories) was used. If the quantitative variables were dependent (because they belonged to the same subject), the paired Wilcoxon test or the Friedman test was used (if there were more than two categories). Values of *p* < 0.05 were considered statistically significant.

## 3. Results

Twenty-nine CALHIV under follow-up in five hospitals in Madrid were included in this study. Their baseline and HIV-related characteristics are presented in Table 1. There were 17 women (58.6%) and 12 men (41.4%). The median age at inclusion was 16.6 years (IQR 11.4–21.6). Among the 29 CALHIV, 68.9% of them were born in Spain, followed by 17.2% from Latin America. The main route of infection was perinatal (22, 75.9%). Two adolescents were sexually infected and in five CALHIV the route of infection was unknown.

Thirteen CALHIV (44.8%) were classified as CDC class A and six (20.7%) as CDC class C. All patients were on ART with a three-drug regimen. This regimen included an integrase inhibitor in 26 (89.7%) of CALHIV. The median CD4+ count at inclusion was 902 cells/mL (IQR: 645–1070). Four CALHIV (13.8%) presented with <500 CD4+/mL and two of them with <250 CD4+/mL. Twenty-six patients (89.7%) presented undetectable viral load. The other three patients presented 1783, 1019, and 52 copies/mL, respectively, at inclusion, and maintained a viral load in serum above 50 copies/mL throughout the study period.

All patients had been previously immunized with a primary series of two doses of the Men C-conjugate vaccine during the first year of life, according to the Spanish Paediatric Association schedule [15]. Median time from the last vaccination dose against Men C was 185 months (IQR 126–234). There were also 11 (37.9%) CALHIV vaccinated with the four-component recombinant protein-based Men B vaccine (4CMenB). The median time elapsed since the last dose of the Men B vaccine was 56.3 months (IQR 31.4–59.1).

All patients included received two doses of the Men ACWY-TT vaccine, a median of 10.9 weeks apart (IQR: 9.1–16.7). There were no adverse events, deaths, or meningococcal infections documented after vaccination. No patient reported any significant adverse event in the first 48 h after vaccine administration or recalled any significant adverse reactions during follow-up in the outpatient clinic. There were 29 sera taken at baseline. At T1, a median of 10 weeks (IQR 6.2–16) after the second dose of the vaccine, 29 sera were collected. Sera at T2 were available only for 24 patients, a median of 58.9 weeks (IQR 54.2–61.8) after the first dose of the vaccine.

At study entry, 44.8% of subjects (N = 13) had seroprotective rSBA titers to at least one serogroup. Individual seroprotection rates at T0, T1, and T2 are shown in Table 2. At T1, all but one child (96.6%, N = 28, CI 95% 89–103) showed seroprotective rSBA titers to the three analyzed serogroups. Twenty children (68.9%) showed a vaccine response to all vaccine serogroups. At T2, 18 out of 24 (75%) had an rSBA titer ≥1:8 to all vaccine serogroups. Again, all but one patient (95.8%, N = 23, CI 95% 87.8–103.8) kept showing seroprotective rSBA titers to serogroups W and Y, but seroprotection rates for serogroup C were 75% (N = 18, CI 95% 57.6–92.3).

### 3.1. Men C Response

The evolution of Men C antibody titers is represented in Figure 1. At baseline, seven (24.1%) had protective antibodies against capsular serogroup C. After vaccination (T1), 24 (82.8%) CALHIV showed a vaccine response. Four patients who had protective antibodies at baseline maintained them at T1 but did not reach a four-fold increase after vaccination. One subject (patient n.10) presented an rSBA titer < 1:8 at this time point.

After 12 months (T2), 18 out of 24 (75%) patients with available samples maintained protective antibody titers. There were five patients who had previously shown a vaccine response in whom no protective antibodies were detected after one year. Patient n.10 maintained titers < 1:8 at T2.

### 3.2. Men W Response

Figure 2 shows antibody evolution against the W capsular serogroup. Eight (24%) CALHIV had protective antibodies at T0. After two vaccine doses, at T1, 24 patients (82.8%) showed a vaccine response. Four CALHIV who had protective antibodies at baseline did not show four-fold-increased titers but they were maintained, and patient n.10 had no vaccine response.

At T2, 23 out of 24 patients with available samples showed protective antibodies, and patient n.10 had no seroprotective antibodies.

### 3.3. Men Y Response

The evolution of antibodies against Y is presented in Figure 3. Nine adolescents (31%) had protective antibodies at T0. After vaccination (T1), 24 (82.8%) showed a vaccine response. There were four patients classified as having no vaccine response, who had protective antibodies but did not reach a four-fold rise in rSBA antibodies. Again, patient n.10 showed an rSBA titer < 1:8.

After one year (T2), all patients but patient n.10 maintained protective antibodies against the Y capsular group (95.8%).

### 3.4. Predictors of Response to Each Serogroup

The HIV-related characteristics of the patients who reached (or did not reach) vaccine response or who maintained protective antibodies against each serogroup are presented in Table 3 and Table 4.

There were three patients whose viral load in plasma was >50 copies/mL at the moment of inclusion. Only one of them presented a vaccine response to Men C at T1. This patient showed a vaccine response to Men C at T1 and presented titers > 1/8 against W and Y at T1 and T2, but titers were not increased four-fold. The other patient presented seroprotective titers against C, W and Y at baseline and at T1 and T2, but they were not increased four-fold. The last one was patient n.10. She was an adolescent who never developed protecting antibody titers against C, W, or Y (neither at T1 nor T2). She was perinatally infected, classified as B-CDC and receiving TAR with poor adherence, presenting a detectable viral load at the time of inclusion and at baseline of 693 CD4+/mm^3^.

Among CALHIV with poor immunological control (<500/mm^3^ CD4+ cell count), two patients showed a vaccine response to Men C and three of them to the W and Y serogroups. At T2, when the sera were analyzed, the two patients who elicited vaccine responses maintained seroprotective titers against the C serogroup. Seroprotection titers against W and Y were also conserved in the three patients who showed vaccine response.

All CALHIV classified as CDC class C showed vaccine responses to all serogroups at T1. When sera at T2 were available, only one patient showed decreased titers against Men C, whereas seroprotective titers against W and Y were maintained one year after vaccination.

Nevertheless, the small sample size limited the analysis of possible risk factors associated with vaccine response. There were no statistically significant differences in viral load, CD4+ count or nadir, CDC classification, or history of Men B vaccination among patients in who a vaccine response was elicited at T1 versus no response. Similarly, none of the studied variables could predict duration of protective antibodies over the follow-up period (T2).

## 4. Discussion

To the best of our knowledge, this is the first study carried out on children and adolescents living with HIV and vaccinated with a two-dose booster schedule of the only Men ACWY-TT vaccine (Nimenrix^®^) available in Spain at the initiation of the study. In our study, most children and adolescents living with HIV showed a strong immunological response and achieved appropriate antibody-mediated protection, as determined by rSBA, against capsular serogroups C, W, and Y. Although most participants maintained protective titers over time, a trend of a decrease in antibody titers for serogroup C was observed over the first 12 months after vaccination. According to our results, vaccines appear to be safe in this population, as previously described in immunocompetent children and in line with previous experiences with other tetravalent conjugated meningococcal vaccines in CALHIV [16,17,18]. Of note, most of our patients were on effective ART, virologically suppressed and with a good immunological and clinical status. Since these factors may influence vaccine response, our results may not be replicable in more disadvantaged contexts or situations.

Interestingly, around one-quarter of the adolescents in our study showed baseline immunity to some of the capsular serogroups analyzed before vaccination. This percentage is lower than that reported by Bermal et al. in immunocompetent adolescents and adults [19]. They described that, prior to vaccination, up to 58–90% of participants had rSBA titers >1:8 against the four capsular groups. Nevertheless, our results are in line with the percentages published by Siberry et al. [17] for HIV-infected adolescents: they found antibody titers against C, W, and Y at baseline in 11%, 15%, and 35% of patients, respectively, although the definition of protection differs. Siberry et al. used 1/128 as the cut-off to define seroprotection titers. Considering this cut-off in our study, 6.9% of patients presented protection against C and 24% against W and Y prior to vaccination.

It is important to highlight that all the children and adolescents included in our study were previously immunized with a primary series of the Men C conjugate vaccine during the first year of life. They received two doses: one at 4 months and another at 12 months of age. However, despite receiving the primary immunization, only a small proportion of CALHIV showed immunity to Men C. This finding is consistent with those of previous studies conducted in immunocompetent children and adolescents, which have also reported low sustained specific antibody responses to Men C [20]. The immunogenicity observed to serogroup C from conjugated vaccines has consistently been shown as having the lowest serogroup response rates across different studies [16,17,18].

Children and adolescents living with HIV are known to have an impaired response to immunizations [21,22]. In our study, with the two-dose schedule, around 83% of CALHIV showed an immune response for each serogroup. Siberry et al. reported similar data in a study using the quadrivalent meningococcal vaccine conjugated with diphtheria, both in 2- to 10-year-old children [16] and in adolescents or young adults [17,18] living with HIV. In these studies, the authors found a vaccine response in 80% of children against the C capsular group, 84% against Y, and 100% against W. Although it is difficult to compare, this observed response is adequate in most patients but lower than that reported in healthy subjects after immunization with the TT-conjugated tetravalent vaccine [23]. Also, a recent study conducted in the UK on individuals living with HIV but older than 20 years of age (median age 36 years old, IQR 20–45), not perinatally infected, with an undetectable viral load in 83% of them, showed that 94% showed protective antibodies against Men C and 100% against Men W and Men Y one month after a two-dose schedule of the quadrivalent Men ACWY vaccine conjugated to the diphtheria CRM197 protein carrier [11].

Regarding specific antibody evolution over time, our data are in keeping with the observations previously reported by other groups, showing that C capsular group antibodies are frequently lost or at least the titer wanes more rapidly. After one year, only 75% of patients maintained protective antibodies against the C serogroup, in contrast with the sustained response observed against the W and Y serogroups (95.8% in both). Our data show a slightly higher seroprotection level at week 52 after vaccination than the one described at 72 weeks by Siberry et al. (45% for C, 95% for W, and 91% for Y) using the diphtheria-conjugated vaccine. This should be interpreted carefully because of the different vaccines administered and the follow-up period the different definition of seroprotection, and the different ages and HIV-related characteristics of the study population.

Our study provides insight into the use of a two-dose Men ACWY-TT (Nimenrix) vaccine schedule in CALHIV, reporting safety and acceptable immunogenicity for all serogroups up to one year after vaccination. It has several limitations, including the lack of a standardized questionnaire to collect adverse events after vaccine administration. Another limitation is related to the relatively small sample size and the follow-up during the COVID-19 pandemic, which precluded or delayed collection of serum samples at T1 and T2 in some patients. Another limitation of our study was that all children and adolescents were on ART and presented a good immuno-virological status, limiting our ability to address factors determining vaccine response. Nevertheless, our study is, to the best of our knowledge, the first to assess the immune response of the quadrivalent Men ACWY-TT vaccine in children and adolescents living with HIV, mostly perinatally infected, showing adequate protection that persists for at least 12 months in most patients. Longer follow-up would be required to define the need for booster doses in this special population.

In summary, children and adolescents living with HIV on effective ART and in a good immuno-virological situation show an appropriate vaccine response after two doses of the Men ACWY-TT vaccine (83% for each serogroup). Moreover, antibody-mediated protection against capsular serogroups C, W, and Y is maintained in more than 70% of patients one year after vaccination.

## Figures and Tables

**Figure 1 microorganisms-12-00030-f001:**
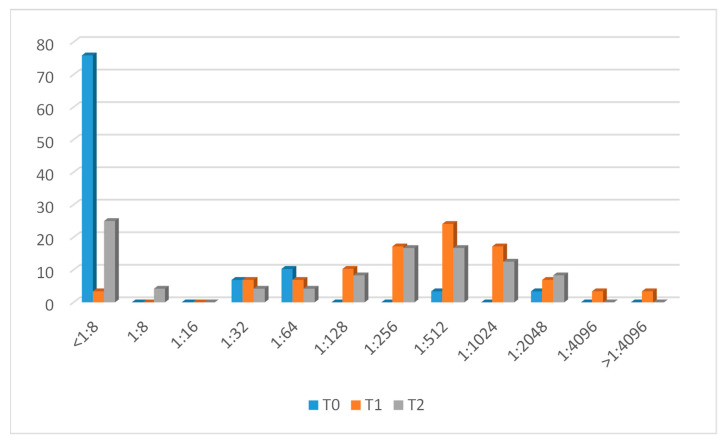
Men C titer evolution.

**Figure 2 microorganisms-12-00030-f002:**
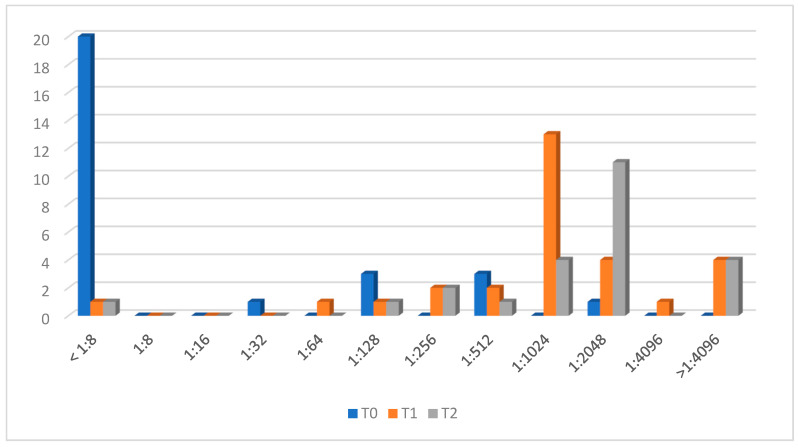
Men W titer evolution.

**Figure 3 microorganisms-12-00030-f003:**
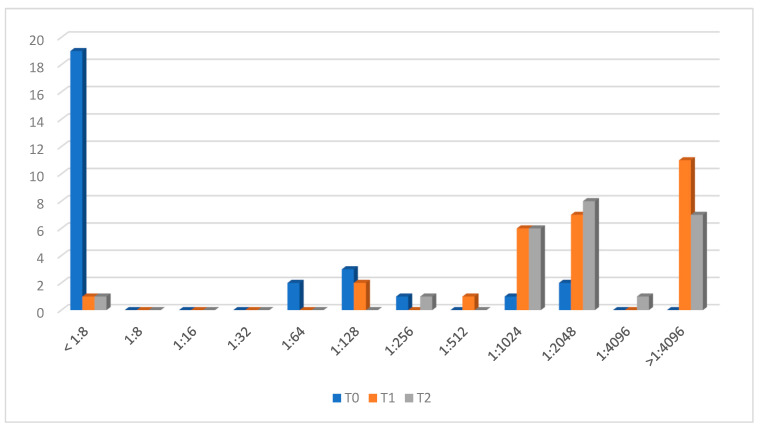
Men Y titer evolution.

**Table 1 microorganisms-12-00030-t001:** CALHIV characteristics.

Total CALHIV N = 29	N (%)
Demographic characteristics	
Sex at birth	
Female	17 (58.6%)
Male	12 (41.4%)
Age	
Median (IQR)	16.6 years (11.4–21.6)
<12 years old	9 (31%)
12–18 years old	11 (38%)
>18 years old	9 (31%)
Place of birth	
Spain	20 (68.9%)
Latin America	5 (17.2%)
Sub-Saharan Africa	3 (10.3%)
India	1 (3.4%)
HIV-related characteristics	
Route of HIV transmission	
Vertical	22 (75.9%)
Sexual	2 (6.9%)
Unknown	5 (17.2%)
CDC classification	
A	13 (44.8%)
B	10 (34.5%)
C	6 (20.7%)
Antiretroviral treatment	
3-drug TAR	29 (100%)
InINT in TAR	26 (89.7%)
Immuno-virological status	
CD4+ cell count	
Nadir, median (IQR)	446 cells/mL (333–644)
CD4+ at inclusion, median (IQR)	902 cells/mL (645–1070)
<500 CD4+/mL at inclusion	4 (13.8%)
<250 CD4+/mL at inclusion	2 (6.9%)
Viral load	
Undetectable (<50 copies/mL serum HIV-RNA)	26 (89.7%)

**Table 2 microorganisms-12-00030-t002:** Seroprotection and vaccine response.

	C	W	Y	All of them
Seroprotective rSBA titers				
T0 (N 29)	7 (24.1%)	8 (27.6%)	9 (31%)	2 (2.9%)
T1 (N 29)	28 (96.6%)	28 (96.6%)	28 (96.6%)	28 (96.6%)
T2 (N 24)	18 (75%)	23 (95.8%)	23 (95.8%)	18 (75%)
Vaccine response				
	24 (82.8%)	24 (82.8%)	24 (82.8%)	20 (68.9%)

rSBA: serum bactericidal antibody assays using rabbit complement.

**Table 3 microorganisms-12-00030-t003:** Characteristics of patients with vaccine responses to different serogroups at T1, N29.

	Men C Vaccine Response		Men W Vaccine Response		Men Y Vaccine Response	
	Ýes, N 24	No	Ýes, N 24	No	Ýes, N 24	No
Viral load						
Detectable, (N 3)	1 (3.4%)	2 (6.9%)	0	3 (10.3%)	0	3 (10.3%)
Undetectable, (N 26)	23 (79.3%)	3 (10.3%)	24 (79.3%)	2 (6.9%)	24 (79.3%)	2 (6.9%)
Immunological status						
<500/mm^3^ CD4+ (N 4)	2 (6.9%)	2 (6.9%)	3 (10.3%)	1 (3.4%)	3 (10.3%)	1 (3.4%)
>500/mm^3^ CD4+ (N 25)	22 (75.8%)	3 (10.3%)	21 (72.4%)	4 (13.7%)	21 (72.4%)	4 (13.7%)
CDC classification						
A (N 13)	9 (31%)	4 (13.7%)	11 (37.9%)	2 (6.9%)	11 (37.9%)	2 (6.9%)
B (N 10)	9 (31%)	1 (3.4%)	7 (24.1%)	3 (10.3%)	7 (24.1%)	3 (10.3%)
C (N 6)	6 (20.6%)	0	6 (20.6%)	0	6 (20.6%)	0

**Table 4 microorganisms-12-00030-t004:** Characteristics of patients with seroprotective antibodies to different serogroups at T2, N24.

	Seroprotection to Men C		Seroprotection to Men W		Seroprotection to Men Y	
	Ýes, N 18	No	Yes, N 23	No	Yes, N 23	No
Viral load						
Detectable (N 3)	2 (8.3%)	1 (4.2%)	2 (8.3%)	1 (4.2%)	2 (8.3%)	1 (4.2%)
Undetectable (N 21)	16 (66.6%)	5 (20.8%)	21 (87.5%)	0	21 (87.5%)	0
Immunological status						
<500/mm^3^ CD4+ (N 4)	2 (8.3%)	2 (8.3%)	3 (12.5%)	1 (4.2%)	3 (12.5%)	1 (4.2%)
>500/mm^3^ CD4+ (N 20)	16 (66.6%)	4 (10.3%)	20 (83.3%)	0	20 (83.3%)	0
CDC classification						
A (N 12)	10 (41.6%)	2 (8.3%)	12 (50%)	0	12 (50%)	0
B (N 8)	5 (20.8%)	3 (12.5%)	7 (29.2%)	1 (4.2%)	7 (29.2%)	1 (4.2%)
C (N 4)	3 (12.5%)	1 (4.2%)	4 (16.6%)	0	4 (16.6%)	0

## Data Availability

Data are contained within the article.

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
