# Peer review of "Immunogenicity of the Conjugate Meningococcal ACWY-TT Vaccine in Children and Adolescents Living with HIV"

_microorganisms, 2023, doi:10.3390/microorganisms12010030_

Round 1

Reviewer 1 Report

Comments and Suggestions for Authors

The paper deals with the serious problem of Meningococcal infection among adolescents and young children living with HIV. It is well known that the fulminant course of meningococcal infection is characterized by the development of meningitis and sepsis - the most severe or generalized forms of the disease, associated with a high level of complications and mortality, especially in immunocompromised people. To date, the most reliable way to protect against Meningococcal infection is vaccination. The authors assessed the immunogenicity of the quadrivalent Men ACWY-TT vaccine in immunocompromised young people and showed that children and adolescents living with HIV, and who were on effective ART with good immune-virological indicators, achieved appropriate vaccine response after two doses of Men ACWY-TT vaccine. Antibody-mediated protection against capsular serogroups C, W and Y was maintained in more than 70% of the patients one year after vaccination. The paper is well written and the results are presented well. Though, there are a few major flaws.

Major Lines 185-186 “HIV-related ones are presented in Error! Reference source not found..”: What does it mean? The same applies to the Line 206, Lines 217-218, Line 229, Line 237. Apparently, there was some kind of error when uploading the file to the system. Line 186 “There were 17 females (58.6%), 68.9% born in Spain and (75.9%) were perinatally infected” - the phrase is not quite clear and requires clarification, as well as data presented in Table 1. It is necessary to give a more detailed information on age of study participants. Also, a more detailed information is needed on route of infection of all females. In total, there were 29 study participants diagnosed with HIV, so 12 were males?       It is necessary to provide data on them, also. A more detailed information is necessary regarding viral load. For better perception, it is advisable to give information on predictors of response to each serogroup, also in the form of a Table.

Minor: It is better to highlight the ethic statement in a separate section.

-        Lines 221, 226, 233, 235 etc.: “One subject (patient 10) presented....” it will be better   to say “patient n.10”.

Author Response

Dear reviewer, thanks in advance for your suggestions which help to enrich our manuscript.

As you mention, problems in lines 185-186, 217-218, 229, 237 are in relation to an error during uploading the file. In those spaces there should appear references to Table 1, Table 2, Figure 1, 2 and 3. Now these errors are corrected, and the text has sense.

Line 186 and the first two paragraphs have been adapted to your suggestions, adding more information of the patients. We tried to clarify all demographics data of the CALHIV (including age information or route of infection) both in the manuscript and in the table 1. Information of the immunovirological situation of the patients has also been improved.

Two new tables (table 3 and 4) have been included with information about predictors of response to each serogroup as you recommended.

We have also changed references to patient 10, following your suggestion. Now this patient is identified as “patient n.10”.

Finally, we agree with your recommendation and ethical considerations have been presented in a separate section (2.3) in the new manuscript. 

Reviewer 2 Report

Comments and Suggestions for Authors

Please consider the following comments:

There are many (Error! Reference source not found) in the manuscript so please modify them.

The rationale of the study should be clarified in the introduction and more recent references should be cited.

A graphical abstract representing the idea of the manuscript should be added.

The g letter of Gram should be a capital letter.

Some sentences are too long and should be rewritten

Conclusion of this research article should be more concise 

Comments on the Quality of English Language

English editing is required

Author Response

Dear Reviewer,

We are really thankful for your suggestions that contribute to improve the quality of our manuscript.

We have already modified all lines in which (Error! Reference source not found) appeared. It was due to an error during the process of uploading the manuscript, but it does not appear now. We are sorry for the inconvenience.

According to your suggestion, the introduction has been review adding more information and references to justify the objective of the study.

We appreciate your grammatical recommendations and G of GRAM has already been modified.

English has been reviewed by a native English speaker, too. Sentences have been rewritten not to be so long.

Finally, conclusions have been reviewed and presented as follows: “In summary, children and adolescents living with HIV on effective ART and good immune-virological situation achieve appropriate vaccine response (83% for each serogroup) after two doses of Men ACWY-TT vaccine. Moreover, antibody-mediated protection against capsular serogroups C, W and Y is maintained in more than 70% of the patients one year after vaccination”

Round 2

Reviewer 2 Report

Comments and Suggestions for Authors

the manuscript could be accepted

Comments on the Quality of English Language

Minor errors are present